# Waters from the Djiboutian Afar: A Review of Strontium Isotopic Composition and a Comparison with Ethiopian Waters and Red Sea Brines

**Tiziano Boschetti** [1], **Mohamed Osman Awaleh** [2] **and Maurizio Barbieri** [3,*]

1   Department of Chemistry, Life Sciences and Environmental Sustainability, University of Parma, Parco Area delle Scienze, 43124 Parma, Italy; tiziano.boschetti@unipr.it

2   Institut des Sciences de la Terre, Centre d'Etudes et de Recherches de Djibouti (CERD), Route de l'aéroport P.O. Box 486, Djibouti–ville, Republic of Djibouti; awaleh@gmail.com

3   Department of Earth Sciences, Sapienza University of Rome, Piazzale A. Moro 5, 00185 Rome, Italy

*   Correspondence: maurizio.barbieri@uniroma1.it; Tel.: +39-06-49914593

**Abstract:** Drinking water is scarce in Djibouti because of the hot desert climate. Moreover, seawater intrusion or fossil saltwater contamination of the limited number of freshwater aquifers due to groundwater overexploitation affect those who live close to the coastline (~ 80% of the population). Despite this, the geothermal potential of the country's plentiful hot springs could resolve the increasing electricity demand. Strontium isotopes ($^{87}Sr/^{86}Sr$) are routinely used to determine sources and mixing relationships in geochemical studies. They have proven to be useful in determining weathering processes and quantifying endmember mixing processes. In this study, we summarise and reinterpret the $^{87}Sr/^{86}Sr$ ratio and Sr concentration data of the groundwater collected to date in the different regions of the Djibouti country, trying to discriminate between the different water sources, to evaluate the water/rock ratio and to compare the data with those coming from the groundwater in the neighbouring Main Ethiopian Rift and the Red Sea bottom brine. New preliminary data from the groundwater of the Hanlé-Gaggadé plains are also presented.

**Keywords:** strontium isotopes; hydrogeochemistry; groundwater; seawater intrusion; mixing processes

## 1. Introduction

The Afar Triangle is the most geologically active area in East Africa, as evidenced by its volcanic rocks, active volcanoes, and hydrothermal manifestations, such as hot springs and fumaroles [1]. In Djibouti, according to the location of the surface manifestations, at least 13 sites for geothermal energy development have been identified, with a global potential of approximatively 1000 MW [2,3]. The most active structure is located in the Asal-Fiale rift, which is the westward prolongation of the Gulf of Aden–Gulf of Tadjoura Ridge (Figure 1). Since the beginning of the 1970s, Asal was the main target area of the geothermal research programs in the country, as the area is the most promising in terms of its high enthalpy deep source with a reservoir temperature of up to approximatively 360 °C [4]. However, due to the high salinity and related scale problems of the Asal fluids, the neighbouring Hanlé-Gaggadé graben area has been considered as an alternative option [5,6] (Figure 1). The Hanlè and the Gaggadé grabens are located in the southwestern region of Djibouti, named Dikhil, mid way between Lake Abhé and the Asal Rift (Figure 1). In this area, basalts of the Stratoid Serie (3.4 Ma) circumscribe the sediment-filled grabens with an interposed rhyolitic centre of volcanic origin (the Babo Alou mountain, Figure 1, [1]). The main structural trends and faults are parallel to those of the Afar Rift systems. Hydrogeological maps and transmissivity data of the main lithological units are available in the literature [7,8].

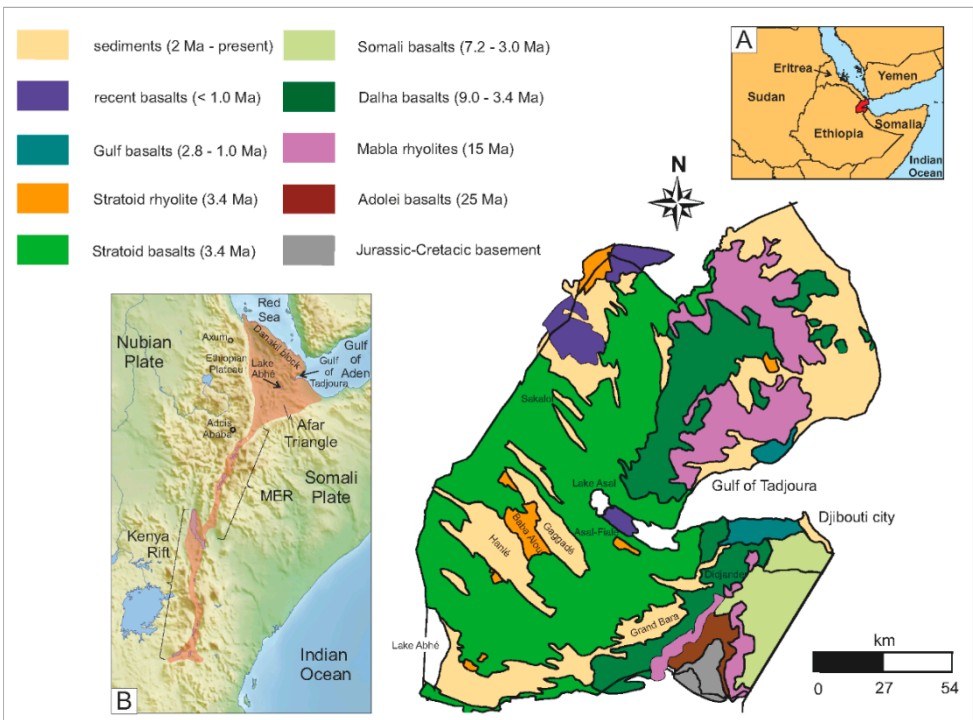

**Figure 1.** A schematic geological map of the Republic of Djibouti (modified after [9]). Inset A: location of the Republic of Djibouti (red) in the Horn of Africa. Inset B: a subdivision of the East African Rift System (light red; modified from [10]). The Hanlé and Gaggadé plains are the location at which the water samples were collected in this study. Details of the location at which the samples were collected are furnished in the Supplementary File S1 (a Google Earth *.kmz file).

The isotope ratio of Sr (expressed as $^{87}$Sr/$^{86}$Sr) has been shown to be an effective tracer of water–rock interactions, and has been used to identify and quantify sources and mixing relationships in hydrogeochemical studies [11–15]. Unlike stable isotopes of oxygen, hydrogen, and carbon, the isotope composition of Sr is not affected by evaporation (e.g., from holding ponds) or biological activity. For the strontium isotopes ($^{87}$Sr/$^{86}$Sr) to be used successfully as a natural tracer in ground and surface waters, the isotope ratios of the potential endmembers must be distinct. If potential endmembers have distinct Sr isotopic ratios, Sr isotopes can be used to identify waters interacting with rocks from specific stratigraphic units [11,14,16–19] and can even be used as sensitive indicators of seawater intrusion [20–22] or minute amounts of contamination [23–25].

In this study, we summarise and reinterpret the main results of the groundwater studies in Djibouti concerning the $^{87}$Sr/$^{86}$Sr data [20,26–29], comparing these with other data from the literature that come from neighbouring areas of active rifting (mainly the Main Ethiopia Rift and the Red Sea), and presenting new preliminary data on groundwater from the Hanlé-Gaggadé plains.

## 2. Geological Background and the Strontium Isotope Ratio of the Rocks

The Cenozoic flood basalts of Djibouti and Ethiopia, along with those from Yemen, pertain to a single petrologic province [30,31]. Structurally, the so-called Afar depression or triangle is a R-R-R-type (ridge-ridge-ridge) triple junction, which was formed by the combination of three major extensional structures: the Red Sea, the East African Rift System (EARS), and the Gulf of Aden (Figure 1). It formed by the rise of the asthenospheric head called the 'Afar Plume', which caused the breakup of the Horn of Africa (i.e., thinning of the lithospheric mantle and continental rifting) and the birth of the associated volcanic activity in the area. Actually, the Afar Triangle is bordered on the west by the Ethiopian Plateau and escarpment, by the Danakil block to the northeast (between it and the Red Sea), by the Somalian Plateau and escarpment to the south, and to the southeast by the Ali-Sabieh block

(adjoining the Somalian Plateau). The $^{87}Sr/^{86}Sr$ signatures of the basalts from the Republic of Djibouti present a large $^{87}Sr/^{86}Sr$ variation between 0.70309 and 0.70670 [32–34]. Despite this, the modern 'Afar Plume' has a restricted range. In the Gulf of Aden, it displays an $^{87}Sr/^{86}Sr$ ratio between 0.70280 and 0.70414 [30], with a central value of approximatively 0.7035, which is widely recognised and accepted [35,36]. The basalts from Adolei, Ali Adde, and Mablas, also characterised by the presence of $^{87}Sr$-enriched rhyolitic rocks, show systematic Sr isotope ratios of $^{87}Sr/^{86}Sr \gg 0.70414$, which is greater than the present-day Afar plume's upper limit. The basalts from the abovementioned localities are the oldest found in Djibouti (from 30 to 10 Ma), and it has been supposed that the modern Afar volcanism is derived from the same plume that fed such Oligocene flood basalts [30,31]. A quite similar range between 0.70318 and 0.70665 was detected in the basalt lavas that initially erupted from the Main Ethiopia Rift (MER, the northeast-trending sector of the EARS, Figure 1), with higher values detected in the rhyolitic products [30].

## 3. Materials and Methods

Water samples from 12 sampling sites in the Hanlé and Gaggadé plains were collected (Figure 1 and the Google Earth map in the Supplementary File S1). The sampling procedure and the physicochemical analysis on site (temperature, pH, electrical conductivity) and in the laboratory (quadrupole ICP-MS determination of Sr concentration) are described in detail elsewhere [20]. Water aliquots dedicated to the determination of the $^{87}Sr/^{86}Sr$ ratio were filtered through 0.45 μm membrane filters (Millipore, Burlington, MA, USA) and acidified to pH < 2 with Suprapur® $HNO_3$ (Merck, Kenilworth, NJ, USA) before being collected in 250 ml polyethene containers. Sr-isotopes were determined by an MC ICP-MS (Nu Plasma II, Nu Instruments, Wrexham, UK) at the GEOTOP laboratory (Montréal, QC, Canada) following an adapted method by Trincherini and colleagues [37]. Calibration was performed following an external intra-sample-standard bracketing using the expected ratio $^{87}Sr/^{86}Sr = 0.71025$ of the reference material NIST SRM 987 (NIST, Gaithersburg, MD, USA). A true $^{86}Sr/^{88}Sr$ ratio of 0.1194 was assumed throughout, with all data corrected for instrumental mass bias using an exponential law [38]. The analytical error on replicates was between ±0.00002 and ±0.00024 (2σ).

## 4. Results

The mean strontium isotope ratio of the waters collected in the Hanlé-Gaggadé plain is 0.70595 ± 0.00070 (Table 1), which is at the central value of all of the previously investigated groundwater and basalts collected to date in Djibouti. Indeed, excluding those samples that were clearly mixed with seawater, the thermal and non-thermal waters from Djibouti studied to date show a strontium isotope ratio range of $0.70367 < {}^{87}Sr/^{86}Sr < 0.70702$ [20,26–29], which is very close to the range of the Djiboutian basalts.

**Table 1.** The coordinates, strontium concentration, isotope ratio, and principal physicochemical properties of the collected water samples.

| Sample Name | Latitude N | Longitude E | Sample Type | * Chemical Facies | T (°C) | pH | * TDS (g/L) | Sr (mg/L) | $^{87}Sr/^{86}Sr$ |
|---|---|---|---|---|---|---|---|---|---|
| Oudgini | 11°30.704′ | 41°56.500′ | spring | Na-Cl | 40 | 8.18 | 1.80 | 0.198 | 0.70635 |
| Agna | 11°34.059′ | 41°54.780′ | spring | Na-Cl | 41 | 8.21 | 1.67 | 0.094 | 0.70629 |
| Minkillé | 11°39.253′ | 41°57.032′ | spring | Na-Cl | 52 | 7.81 | 2.07 | 0.533 | 0.70430 |
| Sâgallé | 11°39.151′ | 41°54.646′ | spring | Na-Cl | 39 | 8.39 | 1.53 | 0.131 | 0.70649 |
| Ease-moydo | 11°40.302′ | 41°53.155′ | spring | Na-Cl | 36 | 8.75 | 2.69 | 0.326 | 0.70634 |
| Daggirou | 11°36.447′ | 41°58.613′ | spring | Na-Cl | 38 | 8.19 | 2.29 | 0.167 | 0.70559 |
| Dahotto | 11°37.531′ | 41°57.068′ | spring | Na-Cl | 40 | 8.01 | 2.46 | 0.202 | 0.70593 |
| Galafi | 11°42.203′ | 41°50.985′ | borehole | Na-HCO₃ | 36 | 7.96 | 0.68 | 0.081 | 0.70488 |
| Hanlé 1 | 11°21.492′ | 42°8.349′ | borehole | Na-HCO₃ | 36 | 7.90 | 0.44 | 0.690 | 0.70641 |
| Hanlé 2 | 11°23.921′ | 42°4.715′ | borehole | Na-HCO₃ | 34 | 8.30 | 0.68 | 0.156 | 0.70656 |
| Daoudaouya | 11°45.623′ | 42°8.385′ | borehole | Mg/Na-HCO₃ | 40 | 7.16 | 0.29 | 0.367 | 0.70623 |
| Mokoyta | 11°27.083′ | 42°15.991′ | well | Na-Cl | 32 | 7.89 | 1.89 | 0.432 | 0.70600 |

\* Inferred from the preliminary data on the main chemical composition; (chemical facies: major cation-anion).

Waters and rocks from the MER and flood basalts of the Ethiopian plateau show similar ranges of $0.70404 < ^{87}Sr/^{86}Sr < 0.70808$ [39–41] and $0.70318 < ^{87}Sr/^{86}Sr < 0.70665$ [30,42], respectively. Both in the Djiboutian Afar and in the MER, some higher values in the groundwater and interacting rocks are due to the involvement of an enriched mantle or crustal component in the source magma [30,31]. In Figure 2, in which strontium concentrations and the related $^{87}Sr/^{86}Sr$ isotope ratios of all sampled waters are plotted, it is easy to show how groundwater of meteoric origin from Djibouti and Ethiopia fall in a common wide strip. The upper limit curve of this strip has the following endmembers: i) rainwater, with an Sr concentration of 0.005 mg/L and a seawater-like $^{87}Sr/^{86}Sr$ ratio of 0.709175 [43,44], typical of an average rain from coastal areas in a monsoon zone [45]; and ii) basalt with an $^{87}Sr/^{86}Sr$ Asal plume signature of 0.7035 and the lowest detected Sr concentration of 100 ppm, typical of less-evolved basalts [46]. The endmembers can be joined by a binary mixing equation, which would be represented by a straight line if the axes of the diagram plotted as Y = 1/Sr versus X = $^{87}Sr/^{86}Sr$. However, in Figure 2, it is shown as a hyperbole due to the log scale on the ordinate axis (semi-log diagram) [47]. The lower limit curve of the strip is traced between a freshwater endmember (TDS < 1 g/L) in the Didjander valley (Sr = 0.56 mg/L; $^{87}Sr/^{86}Sr$ = 0.70603) [26] and the average composition of the Upper Miocene–Pliocene Dahla basalts (Sr = 373 ± 59 ppm; $^{87}Sr/^{86}Sr$ = 0.70411 ± 0.00047) [32–34]. According to Bretzler et al. [39], and differently to surface waters that have a restricted field (the MER lake and rivers in Figure 2, [39,40]), the wide binary mixing strip for groundwater of meteoric origin could be related to their different Sr concentration, which is characterised by the non-conservative behaviour of the dissolved Sr due to sorption, ion exchange, or dissolution/precipitation reactions [39]. A typical example of calcite precipitation from oversaturated water is from Lake Abhé, as evidenced by i) the presence of wide travertine carbonate chimneys in its surroundings [48], and ii) the related depletion of dissolved Sr concentration until 0.003 mg/L from the mixing strip (1/Sr = 333 in Figure 2). Differently, the most probable maximum concentrations of dissolved strontium in the non-thermal groundwater by interaction with carbonate and sulfate minerals were calculated by a PHREEQCI code (United States Geological Survey, Reston, VA, USA) [49]. In the first modelling, fluid with Sr = 0.195 mg/L was obtained from an equilibrium between pure water at a temperature of 35 °C, carbon dioxide at an atmospheric fugacity of logfCO₂ = −3.41, and a mineral reactant constituted by a solid solution of aragonite and strontianite (this latter mineral was kept at the typical miscibility gap of 0.005 mol) [49]. In the second modelling, the obtained fluid with Sr = 0.447 mg/L resulted from a similar equilibrium in which the solid reactant was substituted with a mineral solid solution between anhydrite and celestite. This latter mineral was kept at 0.005 mol according to a Sr concentration in anhydrite of approximatively 5000 ppm, in line with the maximum Sr concentration in seawater [50] or with its heating at T > 200 °C [51]. The reciprocal of the strontium concentrations obtained by the two above-described models are 5 and 2, respectively. Indeed, Sr-enriched groundwaters from

the Ethiopian MER are grouped within the two horizontal lines in Figure 2 representing the two above-described concentrations.

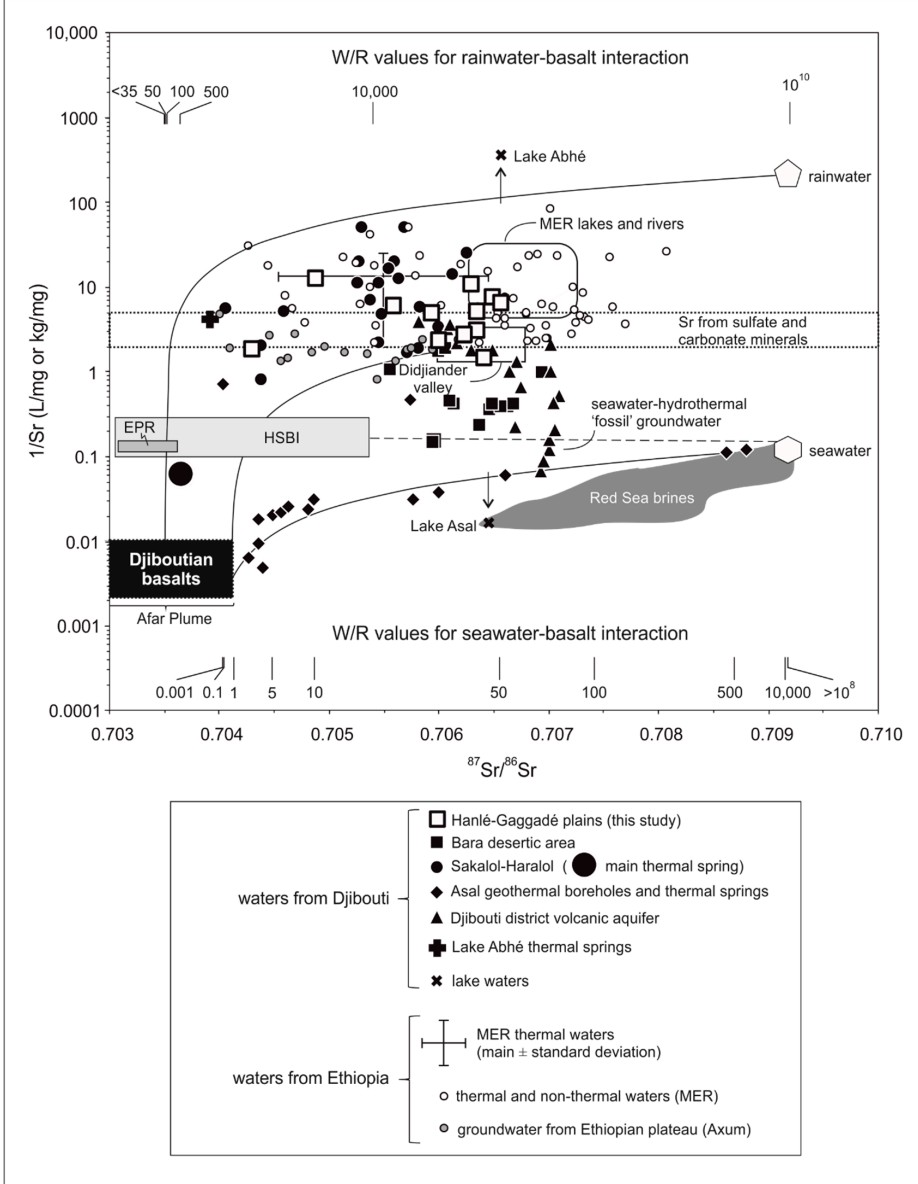

**Figure 2.** A 1/Sr versus $^{87}Sr/^{86}Sr$ diagram showing the comparison between the waters from Djiboutian Afar [20,26–29], the Ethiopian MER [39,40], and the Ethiopian Northern plateau [41]. Solid curves depict both mixing lines between the two endmembers located at the start and the end of each curve [47] and the water–basalt interaction modelling according to Equation (3) [16,52,53]. In this latter case, the obtained *W/R* ratios (*N* in Equation (3)) considering rainwater or seawater as the initial fluid (that is, the uppermost and the lowermost curve, respectively) are also shown. The compositional range of the most recent Djiboutian basalts of the Dahla, Stratoid, and Asal Series (i.e., from 9 Ma to date [30]) and Afar plume represent the main solid endmember or interactant (see the text for details and for the specific compositional range). The hydrothermal waters sampled at the bottom of the Red Sea (the 'Red Sea brines' grey field, [54–58]), those issuing from the 21° N East Pacific Rise (EPR, dark grey field, [59–61]), and those obtained from the Hydrothermal Seawater Basalt Interaction (HSBI, light grey field, [62,63]) are also shown for comparison. The horizontal dashed line depicts the binary mixing between seawater and the HSBI field.

Furthermore, freshwaters both from the Didjander valley and the Djibouti aquifer (Sr = 0.56 ± 0.26 mg/L; $^{87}$Sr/$^{86}$Sr = 0.70639 ± 0.00040) [20] showed a stoichiometric undersaturation [64] in those carbonatic and sulfatic solid-solutions; that is, a tendency to dissolve them (Figure 2). This process could also explain the chemistry of Axum groundwater from the Ethiopian plateau, which has Ca (Mg) as its dominant cations instead of Na as in most of the groundwater from the MER [41,65]. However, other processes, such as mixing between aquifers and bivalent cation adsorption/desorption by clays, could occur and concur to shift the samples away from the two horizontal lines describing the solid-solutions. Most of the waters from the Hanlé-Gaggadé plains collected in this study, including four freshwaters of Na-HCO$_3$ composition from explorative boreholes (Hanlé 1, Hanlé 2, Galafi, and Daoudaouya), fall between the 'Didjander' and 'MER lake and rivers' fields (Figure 2), evidencing their meteoric origin. According to the previous prospections in this area, the low salinity bicarbonate waters, probably fed by the local intermittent floods (wadis), represent the most interesting source for drinking purposes [66,67]. Differently, hot springs with a Na-Cl composition from the same area should be more suitable for geothermal purposes.

## 5. Discussion

### 5.1. Geothermal Waters and the Water/Rock Ratio

The geothermal waters of meteoric origin from Djiboutian Afar and MER fall within the above-described strip. The hot springs from Lake Abhé [28] and Sakalol [27] in Djibouti seem to be 'more evolved'; that is, more shifted towards the $^{87}$Sr/$^{86}$Sr ratios of the interacting rocks (Figure 2) than geothermal waters from the MER [39,40], which have the followings mean values: Sr = 0.14 ± 0.11 mg/L; $^{87}$Sr/$^{86}$Sr = 0.70550 ± 0.00018 [39,40]. The approaching of a water sample towards the field of interacting rock is mainly due to two variables: temperature and the water/rock ratio during the water–rock interaction process. The water reservoirs of Lake Abhé and the Sakalol hot springs have reached a full equilibrium with the basalts of the Stratoid Serie (Sr = 366 ± 68 ppm; $^{87}$Sr/$^{86}$Sr = 0.70366 ± 0.00018 [33,34,68]) at a temperature of 135 °C and 143 °C, respectively [27,28]. The hottest Na-Cl spring water from the Hanlé plain (Minkillé) has probably followed a similar evolution, at least in terms of temperature at depth and interacting rocks [27]. This is also evidenced by its significantly lower $^{87}$Sr/$^{86}$Sr ratio of 0.70430 ± 0.00004 in comparison with the other waters sampled in the same area (Table 1).

Before modelling the water–rock interaction, it should be underlined that both mass-spectrometer-induced and natural fractionation effects on the $^{87}$Sr/$^{86}$Sr ratio of the samples are negated by normalisation to the internationally accepted $^{86}$Sr/$^{88}$Sr = 0.1194 during measurements [69–72]. Therefore, assuming, in a first approximation, that the water–rock fractionation factor on the $^{86}$Sr/$^{88}$Sr ratios is negligible ($\Delta_{w-r} = 0$), the water–rock interaction modelling can be calculated exclusively by the function of the water/rock ratio. In a closed system, the most-used model foresees the following mass balance [73,74]:

$$C_w W \delta_w^i + C_r R \delta_r^i = C_w W \delta_w^f + C_r R \delta_r^f \tag{1}$$

where $i$ = initial value; $f$ = final value; $W$ and $R$ = the mass of water and rock involved, respectively; and $\delta$ = isotope ratio of the element in per mil. For the purpose of calculation, the original $^{87}$Sr/$^{86}$Sr values can be temporally converted to the per mil notation by the following equation: $[(^{87}$Sr/$^{86}$Sr)/0.7047 − 1] × 1000, where 0.7047 is the bulk-earth value [75]. $C_w$ and $C_r$ are the concentrations of the element in water and rock, respectively. Generally, for strontium and the isotope ratios of other elements, only initial concentrations are considered [60,74].

After grouping, substituting $W/R = N$, and rearranging the equation to make the final isotope composition of the water the subject, we have [16,52,53]:

$$\delta_w^f = [N \delta_w^i + (C_r/C_w) \delta_r^i + (C_r/C_w) (\Delta_{w-r})] / [N + (C_r/C_w)] \tag{2}$$

Considering the previous assumption on the $^{87}$Sr/$^{86}$Sr fractionation factor ($\Delta_{w-r} = \delta_w^f - \delta_r^f = 0$), the above equation becomes reduced into the following form:

$$\delta_w^f = [N\,\delta_w^i + (C_r/C_w)\,\delta_r^i]/[N + (C_r/C_w)] \tag{3}$$

The different strontium concentrations and isotope ratios of the basalts and water endmembers employed in the equation model are described in the Supplementary File S2. For an interaction between meteoric water and the Afar plume source rock, the obtained $^{87}$Sr/$^{86}$Sr of the fluid at different *N* values are shown in Figure 2. The hot springs from Lake Abhé and the hottest from Sakalol have an estimated *W/R* ratio of approximatively 1500 and 600, respectively (Equation (3) and Figure 2). Differently, the Asal geothermal waters [29] are mainly displaced between seawater and the Asal basalts (which have the following values: Sr = 359 ± 80 ppm; $^{87}$Sr/$^{86}$Sr = 0.70376 ± 0.00027 [32–34,68,76]), thus confirming their marine origin (Figure 2). Using the same approach, the obtained *W/R* ratios of the deep geothermal boreholes (between 5 and 1) for the hot springs (up to 50) agree with those previously estimated by a different method [77].

## 5.2. Hydrothermal Waters of Seawater Origin and Red Sea Bottom Brines

It should be noted that, because of Sr sequestration by secondary mineral phases (e.g., anhydrite or plagioclase/epidote recrystallisation [78,79]), the seawater–basalt interaction during hydrothermal processes can produce fluids out of the simple binary mixing curve between basalt and seawater endmembers. This was also demonstrated by experimental interactions between hot seawater and basalts, whose produced fluids are displaced on a mixing line between seawater and a hydrothermal fluid with the following composition [62,63]: Sr = 6.1 ± 2.7 mg/L; $^{87}$Sr/$^{86}$Sr = 0.70421 ± 0.00115 (HSBI in Figure 2). The strontium isotope composition of the hydrothermal solution as black smokers discharged from axial mid-oceanic vents falls within this range (e.g., 21° N East Pacific Rise, [59–61]): Sr = 7.6 ± 0.9 mg/L; $^{87}$Sr/$^{86}$Sr = 0.70334 ± 0.00027 (EPR in Figure 2).

Therefore, water samples with a mixed meteoric–marine origin or derived from a hydrothermal seawater–basalt interaction could fall in the area inscribed between the above-explained mixing line and the 'normal' seawater–basalt curve. A typical example is the deep fossil saline groundwater in the complex volcanic aquifer of the Djibouti District [20] (Figure 2). On the opposite side of the seawater–basalt mixing curve, there is brine from Lake Asal (Figure 2), which is located in the Asal-Fiale geothermal district (Figure 1). It showed $^{87}$Sr/$^{86}$Sr = 0.70645 [29], which is at the middle point between seawater and the local basalt composition, thus confirming that the bottom lake is fed by an evolved seawater from the ridge fractures [29,50]. Therefore, the origin seems to be quite similar to the neighbouring Asal geothermal fluid. However, because of the intense evaporation, Lake Asal has a very high salinity (~ 350 g/L, which is approximatively 10 times that of seawater) and a high strontium concentration (Sr = 61 ± 7 mg/L) [29,80]. The reciprocal value of 1/Sr = 0.016 places the representative point of Lake Asal below the seawater–basalt curve (Figure 2). The seawater-derived geothermal aquifer of the Asal-Fiale area is located in an exposed and actively extending continental rift that will become an arm of the Red Sea [50]. In our opinion, Lake Asal is eligible as an experimental site to study the evolution of the deep brines of the Red Sea. Indeed, by observing the strontium composition of the Red Sea brines (Figure 2) [54–58], it is quite easy to deduce how these salt waters found at the bottom of the Red Sea could have evolved in a similar way to Lake Asal, which ideally represents their compositional extreme (Figure 2) [27]. A hydrological model supposes a density-driven migration of the brine from the Afar Depression to the Red Sea [81]. This further supports the evolutionary connection between the two systems.

## 6. Conclusions

This study on the strontium isotope ratio from Afar waters clearly depicts the wide water source variability of the Djiboutian groundwater in comparison with Ethiopian waters. In particular: (i) geothermal waters of Djibouti seem to be more evolved in terms of water–rock interaction, and (ii) coastal aquifers of Djibouti could be affected by the contamination of deep fossil water and seawater intrusion. Furthermore, the comparison of the $^{87}Sr/^{86}Sr$ ratios and strontium concentrations of the brine waters from Lake Asal and the Red Sea bottom show that Lake Asal is eligible as an analogue site to study the evolution of these hydrothermal fluids of seawater origin in the Red Sea rift. Finally, the new preliminary data on groundwater from the Hanlé-Gaggadé plains show that this is an area of interest both for geothermal energy and drinking water purposes. The hottest Minkillé brackish NaCl spring might be a valid low-enthalpy (T < 150 °C) alternative to the salt water of the Asal-Fiale geothermal area, whereas $NaHCO_3$ freshwaters (Hanlé 1, Hanlé 2, Galafi, and Daoudaouya) are better suited for drinking. Indeed, in terms of strontium isotope composition, the former appears as the most evolved in terms of water–rock interaction, whereas the latter are substantially unevolved and indistinguishable from other bicarbonate waters of the country (i.e., the Didjander valley). The $^{87}Sr/^{86}Sr$ isotopic approach used in this (and previous, [20]) studies was remarkably helpful to determine the sources and movements of geothermal and saline groundwater in a coastal aquifer, even in the absence of detailed subsurface geology. The principal findings can aid researchers and local water managers in mapping the subsurface, in developing a conceptual hydrogeologic framework, and in planning for additional groundwater management for drinking water purposes.

**Supplementary Materials:** The following are available online at http://www.mdpi.com/2073-4441/10/11/1700/s1.

**Author Contributions:** Conceptualization, M.B., T.B., and M.O.A.; data curation, T.B. and M.O.A.; investigation, T.B. and M.O.A.; methodology, M.B. and T.B.; software, T.B.; supervision and validation, M.B., T.B., and M.O.A.; writing—original draft preparation, T.B. and M.O.A.; writing—review and editing, M.B. and T.B.

**Funding:** This research received no external funding.

**Acknowledgments:** We would like to thank the two anonymous reviewers for their suggestions and comments.

**Conflicts of Interest:** The authors declare no conflict of interest.

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
