# Peer review of "Waters from the Djiboutian Afar: A Review of Strontium Isotopic Composition and a Comparison with Ethiopian Waters and Red Sea Brines"

_water, doi:10.3390/w10111700_

Reviewer 1 Report

Dear Authors,

I like your work very much. Please concern some suggestions and correction:

I was not able to open file S1 and due to this reason maybe my suggestion in maybe unreasonable.On Fig.1. you can mark sampling locations or add locations on additional map.

Also, in Conclusions somewhere between rows 245 to 247 add few words or sentences about hypsometrical relations and groundwater pathways or design small dimension profile or put some arrows on Fig.1. for readers whose are not familiar with geology and hydrogeology of study region.

Small corrections:

row 76 superscript of atomic mass of Sr

row 93 subscript 3 in formula of acid

row 98 same as 76

row 200 ....water and rock, respectively.

row 205 in bracket normal scription

row 260 subscript 3

Best regards, 

Author Response

C: I was not able to open file S1 and due to this reason maybe my suggestion in maybe unreasonable. On Fig.1. you can mark sampling locations or add locations on additional map.

R: thank you for this comment. Due to its large scale, it is difficult to insert in Fig. 1 the new sample points collected in the Hanlé-Gaggadé plain. Therefore, we have enclosed to the manuscript the S1 file as supplementary material. It is a Google Earth File (.kmz) with the placemarks of the sample points, their names and latitudinal and longitudinal coordinates of the locations. KMZ files can be opened by Google Earth, Google Chrome, Google Drive (https://chrome.google.com/webstore/detail/kml-kmz-viewer-with-drive/mbolhellljccdahaeelobbojpfdgjgco) or GPS Visualizer (a free software, available at http://www.gpsvisualizer.com/). We think that this kind of file is more versatile than a classic gif/jpeg map, because the interested reader could quickly upload and open the file in their smart-tools, in particular if they are interested to visit and/or resample these waters.

C:…in Conclusions somewhere between rows 245 to 247 add few words or sentences about hypsometrical relations and groundwater pathways or design small dimension profile or put some arrows on Fig.1. for readers whose are not familiar with geology and hydrogeology of study region.

R: Unfortunately, a complete hydrological map covering all of the Republic of Djibouti is not available yet. At the time when we write, only partial or local hydrological study are available (e.g. please see hydrological paragraphs in the Awaleh et al. references cited in the text and However, it is in course a big hydrological project by British Geological Survey on Africa, including the Republic of Djibouti with some preliminary maps. In the uploaded revised version, we have cited that project along with a paper on transmissivity data of the geological units (lines 45-46, refs 7 and 8).

Reviewer 2 Report

General Comments

 This is an interesting manuscript largely reviewing (although with a few primary data) 87Sr/86Sr data for groundwaters in Djibouti and some parts of Ethiopia. The scientific discussion is generally well-informed and plausible, although more should probably be made of the ambiguity of determining processes controlling 1/Sr vs 87Sr/86Sr data where these can be determined to some extent both by mixing and evaporative processes (the latter obviously for 1/Sr) - a more detailed discussion of that is required. It is appears that much of the data reviewed has been generated by the authors themselves, so it is critical that the manuscript be modified to indicate how this manuscript is different and builds upon any interpretations previously published by the group. Is tis just an incremental contribution adding a few more data points to the many already published ? Lastly, although the ms is generally understandable and well written, there is significant English language editing to be done.

Recommendation - reconsider after moderate revisions satisfactorily the addressing the general comments above and the specific comments below.

Specific Comments

Title. A “review of” rather than a “review on”

Line 37. “1970s” not “seventies”.

Figure 1. What is the source of the geological information presented ? What is the source of the map on the left ?

Line 50. “samples” not “sample”.

Line 61. “summarise” not “resume”

Table 1. A “,” is used a a decimal point, whereas in the text a decimal point is used as a decimal point. Please replace the “,” in this table as appropriate with a decimal point or consistency with the rest of the text and with normal practice in English-language journals.

Line 139-140. I can guess what is meant here regarding aragonite and dstrontianite but it is not expressed clearly.

Line 146. The reciprocal of a concentration is dependent upon the concentration unit and therefore should be expressed in the appropriate unit.

Line 146-147. This is patently incorrect as many of the “groundwater from Ethiopian plateau (Axum)” data lie outside of the two lines indicated.

Lines 189-191. The normalisation process should be explained, as should the justification for ignoring any possible temperature fractionation effect of 87Sr/86Sr.

Line 193 and throughout. the authors refer frequently to a “water/rock ratio” – in the context of this manuscript, it would be more appropriate to refer to a “water/rock reaction ratio”

Line 199. Subscripts missing for w and r in Cw and Cr respectively.

Line 205. What is the theoretical justification for assuming that the final 87Sr/86Sr ratios of waters and rocks are identical ?

Figure 2. It is unclear how relevant the 87Sr/86Sr values for EPR and HSBI are to the east African system being modelled here.

Author Response

C:Line 139-140. I can guess what is meant here regarding aragonite and dstrontianite but it is not expressed clearly.

R:These modeling have been calculated in order to infer the maximum Sr concentration by interaction of groundwater with carbonate and sulfate minerals (as specified at lines 139-140). The interested reader could obtain all details on the Phreeqci modeling at ref. 48 (the Phreeqci manual). Indeed, we have used the same solid-solution approach listed in the example of that manual. Moreover, we have specified in the revised version that the miscibility gap of 0.005 mol is “typical” on this kind of solid-solution (line 143).

C:Line 146-147. This is patently incorrect as many of the “groundwater from Ethiopian plateau (Axum)” data lie outside of the two lines indicated.

R:Yes, indeed we have written that the process “could” explain the chemical Ca(Mg)-composition. However, after this process, waters could be affected by mixing between two or more component and other processes (as bivalent cation adsorption by clay minerals), which can deviate the samples from the modeled lines (lines 171-173).

C:Line 205. What is the theoretical justification for assuming that the final 87Sr/86Sr ratios of waters and rocks are identical ?

R:That full (or complete) water-rock equilibria has been reached.

C:Lines 189-191. The normalisation process should be explained, as should the justification for ignoring any possible temperature fractionation effect of 87Sr/86Sr

R:The normalisation of the 87Sr/86Sr ratio measurements was proposed just after that “modern” mass spectrometer analyses started. The rationale behind this process is that measured (raw) 87Sr/86Sr values differ from the true values because of fractionation in the mass spectrometer. The measured ratio of other elements (e.g. Ca, Nd, Pb, Hf, Os…) are also corrected for fractionation that take places during the analysis in the mass spectrometer. Unfortunately, the normalization procedure also obliterate the natural fractionation. However, and paradoxically, this also made the success of 87Sr/86Sr isotope ratio, which is used as an “isotope tracer”. We report in the following paragraph a brief desciption of this procedure.

The 86Sr/88Sr = 0.11940 was originally proposed by Nier (1938; now added in the manuscript as reference 71), and accepted by Subcommission on Geochronology in the International Union of Geological Sciences (IUGS). After that, the first internationally accepted normalization procedure multiplied the measured 87Sr/86Sr ratios by a factor f = 2M/(M+0.1194), where M = measured 86Sr/88Sr (Faure and Hurley 1963). Mass difference between 87Sr and 86Sr is only one half that between 86Sr and 88Sr. However, this easy approach improve the within-run precision from ca. 1% to better than 0.001%. For this reason, it was adopted by all investigators worldwide and has been used ever since. More recently, laboratories use an exponential law approach originally proposed for Ca isotopes (Russel et al, 1978; Hart and Zindler, 1989). Measurements made at Geotop and reported in Tab.1 used this approach (as specified at line 102, in the Material and Methods section 3); therefore, we have inserted the citation of the original paper of Russel et al. (1978) in the revised version (ref. 38, line 102). 

Many paragraph/chapters on this argument were written in the isotope geochemistry books used in the degree courses. Therefore, for sake of brevity, we have cited three of them at lines (refs. 68-70). Anyway, in the revised version, we have specified at line 195-198 that normalisation is necessary due to the mass spectrometer-induced fractionation.

C:Line 193 and throughout. the authors refer frequently to a “water/rock ratio” – in the context of this manuscript, it would be more appropriate to refer to a “water/rock reaction ratio”

R:it is implicit that we are talking about water-rock interaction, indeed the term “interaction” is used 22 times in the manuscript. Therefore, we think that it would be a bit pleonastic and redundant to specify it another time. Moreover, equation (2) concerns an estimation of the water and rock globally involved and independently by a specific chemical reaction, which could involve a limited numbers of minerals as solid interactants. 

C:Figure 2. It is unclear how relevant the 87Sr/86Sr values for EPR and HSBI are to the east African system being modelled here.

R:Why not? The EPR and HSBI are both a “seawater-basalt” interaction system, that is two analogues of the Red Sea/Aden rifts (two branches of the R-R-R Afar rift system). 

Above-cited references not included in the references list of the manuscript

Faure, G., Hurley, P. M. (1963). The isotopic composition of strontium in oceanic and continental basalts: application to the origin of igneous rocks. Journal of petrology, 4(1), 31-50.

Hart, S.R., Zindler, A. (1989). Isotope fractionation laws: a test using calcium. International Journal of Mass Spectrometry and Ion Processes, 89(2-3), 287-301.